# A Random Controlled Trial to Examine the Efficacy of Blank Slate: A Novel Spaced Retrieval Tool with Real-Time Learning Analytics

Douglas McHugh [1,*], Richard Feinn [1], Jeff McIlvenna [2] and Matt Trevithick [3,4]

[1] Frank H. Netter MD School of Medicine, Quinnipiac University, Hamden, CT 06518, USA; richard.feinn@quinnipiac.edu
[2] English Language Institute, Pace University, New York, NY 10038, USA; jmcilvenna@pace.edu
[3] Jackson Institute for Global Affairs, Yale University, New Haven, CT 06520, USA; matt@blankslatetech.co
[4] Blank Slate Technologies LLC, Arlington, VA 22203, USA
*   Correspondence: douglas.mchugh@quinnipiac.edu

**Abstract:** Learner-centered coaching and feedback are relevant to various educational contexts. Spaced retrieval enhances long-term knowledge retention. We examined the efficacy of Blank Slate, a novel spaced retrieval software application, to promote learning and prevent forgetting, while gathering and analyzing data in the background about learners' performance. A total of 93 students from 6 universities in the United States were assigned randomly to control, sequential or algorithm conditions. Participants watched a video on the Republic of Georgia before taking a 60 multiple-choice-question assessment. Sequential (non-spaced retrieval) and algorithm (spaced retrieval) groups had access to Blank Slate and 60 digital cards. The algorithm group reviewed subsets of cards daily based on previous individual performance. The sequential group reviewed all 60 cards daily. All 93 participants were re-assessed 4 weeks later. Sequential and algorithm groups were significantly different from the control group but not from each other with regard to after and delta scores. Blank Slate prevented anticipated forgetting; authentic learning improvement and retention happened instead, with spaced retrieval incurring one-third of the time investment experienced by non-spaced retrieval. Embedded analytics allowed for real-time monitoring of learning progress that could form the basis of helpful feedback to learners for self-directed learning and educators for coaching.

**Keywords:** technology enhanced learning; educational software; learning analytics; self-directed learning; formative feedback; educational coaching; spaced retrieval; spaced learning

## 1. Introduction

The human experience of forgetting after initial learning is near ubiquitous [1–4]. It is difficult and effortful for us to assimilate extensive knowledge into long-term memory. For example, health professionals must learn a large volume of information that will be applied months or years later in caring for patients in various clinical circumstances. Regrettably, after initial learning and assessment, knowledge retention by these learners in many instances decays and information is forgotten [5–8]. Human memory and power of recall deteriorates rapidly if we do not reinforce what we have learned.

Spaced retrieval practice involves bringing information from long-term memory back into working memory and enhances retention and knowledge application in both child and adult learners [9–12]. It is a learning technique that incorporates intervals of time between reviews of previously learned material. This is performed in order to exploit the phenomenon whereby humans more easily remember or learn items when they are studied recurrently over a long time span [9,13]. The principle is useful in many circumstances, but especially so where a learner must acquire a large amount of information and retain it indefinitely in memory (e.g., a second language, medical pharmacology, first responder

training and regulations, operational risk and compliance training). Most of us have experienced this in how we learned multiplication tables in mathematics (e.g., $6 \times 7 = 42$, $6 \times 8 = 48$, $6 \times 9 = 54$); we practiced them at intervals again and again until the answers were automatic. In short, educational encounters that are distributed and repeated over time result in more efficient learning and improved retention [14].

Increasing attention is being paid to digital technologies that can enhance existing practices in education, teaching, and learning [15–17]. As a consequence, technology enhanced learning has emerged as a term that describes the use of hardware devices or software applications to support and help learning beyond what might otherwise be readily achieved [18]. This is consistent with the growing desire in higher education settings for learner-centered coaching and helpful feedback to complement self-regulated learning (i.e., self-monitored activities, practices, and behaviors that learners engage in to pursue academic mastery [19–25].

These three contexts prompted Blank Slate Technologies LLC to develop a novel internet-based, spaced retrieval software application with embedded learning analytics called Blank Slate [26]. This was performed with the hope of helping learners keep critical information front of mind (i.e., sustain recurrent successful memory retrievals long term) and for their Total Knowledge Analytics™ platform to allow educators or organizational leaders see who knows what in real time.

The potential implications of this study for teaching and learning practice center around: learners themselves; learners in one-on-one teaching relationships with educators; and learners as part of a learning community (e.g., school class, university course, and work training program). For example, physical therapy, physicians, and sports medicine trainees are among those required to learn the muscles of the human posterior shoulder (i.e., supraspinatus, infraspinatus, teres minor, teres major, subscapularis, deltoid, trapezius, rhomboid major, rhomboid minor, latissimus dorsi, and levator scapulae; Figure 1A). This is an information-dense task that requires learning specific muscle names coupled to their visual identification. It is also something that many people do not find easy and struggle to do without making errors or simply forgetting. Blank Slate provides a means for educators to create 10 digital cards that incorporate the image represented in Figure 1B, each asking learners to name one by one, card by card, muscles A–J. Additionally, educators could create 10 more digital cards using Figure 1B that instead name each specific muscle one by one, card by card, and ask learners to identify the muscle's position A–J on the human back and shoulder. This gives learners practice with two related but distinct retrieval pathways: (1) given the visual position of the muscle, retrieve its name from long-term to working memory, and (2) given the name of the muscle, retrieve its correct location on the back/shoulder from long-term to working memory. This is an example of how spaced retrieval helps to reinforce schema formation by solidifying cognitive frameworks individual learners form when interacting with the material [27]. Rather than learners testing themselves daily (non-spaced retrieval) with these digital cards, Blank Slate has learners test themselves on fewer days that are spaced out across time (spaced retrieval) (Figure 1C). After each question, learners rate the card as easy to remember, hard to remember, or that they had forgotten the information it asked for. Learners would see all 20 cards on day 1. However, thereafter, Blank Slate's learning analytics algorithm would use each individual learner's easy, hard, and forgot ratings to shuffle the cards and present harder cards (i.e., those that they need more practice with) at shorter time intervals and easier cards (i.e., those that they need less practice with) at longer time intervals. This is personalized for each individual learner. Blank Slate's learning analytics can show individual students and educators where they have knowledge gaps and learning deficits; allowing them to adapt what or perhaps how they study, to receive additional teaching or be directed to supplemental resources, or to use learning progress tracking (i.e., seeing forget/hard cards become easier and eventually effortless to recall) to boost learner confidence and raise perceptions of self-efficacy. Lastly, the learning analytics provide data on whole learning cohorts, which can guide educators in what or how they teach (e.g., to

support just-in-time teaching [28] or flipped classroom educational strategies [29–32]). If a large group of learners are struggling with the rotator cuff muscles (i.e., supraspinatus, infraspinatus, teres minor, and subscapularis), educators may adapt their teaching materials and methods in the present (or in future iterations of the class) to focus on these muscles in particular (e.g., by incorporating a case of a patient with a rotator cuff muscle tear from doing bench press exercises poorly in the gym, and having learners speculate over which everyday movements may be impeded and why).

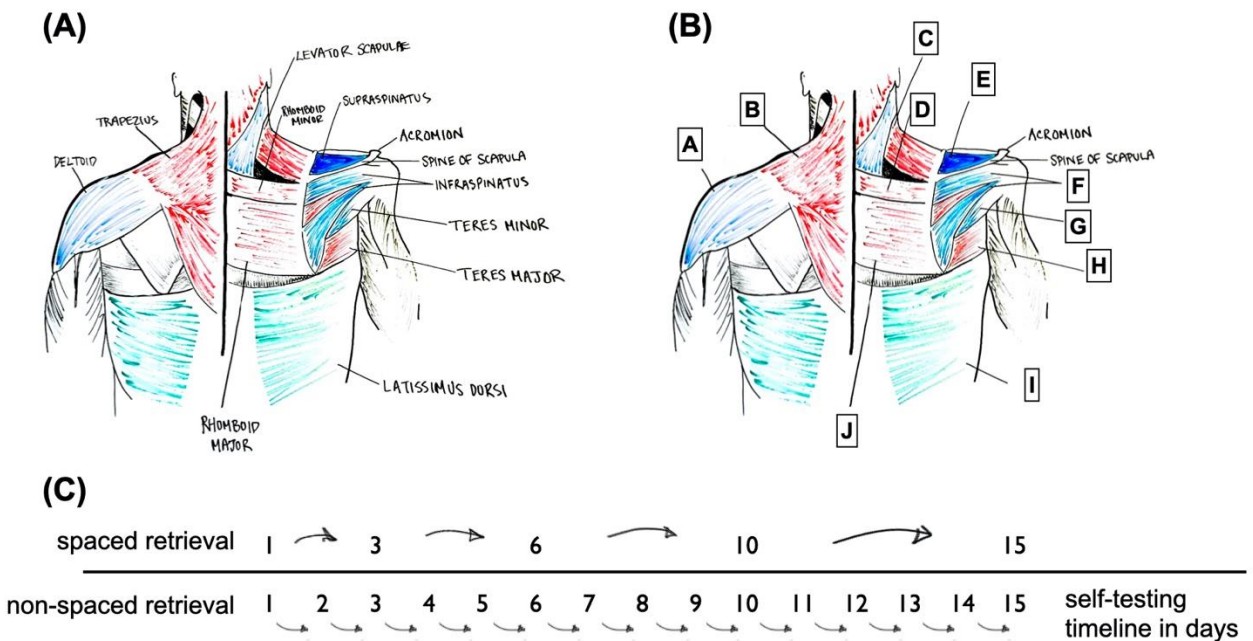

**Figure 1.** Drawing of the human posterior shoulder and back with muscles (**A**) identified and (**B**) de-identified. (**C**) compares example spaced versus non-spaced retrieval motifs across time.

Our research objective was to examine the efficacy of Blank Slate to (1) offset normal human forgetting; (2) unobtrusively monitor learner progress; and (3) create a detailed data record, computationally analyzed to display helpful feedback on individual learner performance. We hypothesized three outcomes: (i) the control group, having no access to Blank Slate, would experience forgetting as measured by post-intervention assessment; (ii) the sequential and algorithm groups would experience increased knowledge acquisition and retention as measured by post-intervention assessment, and (iii) the algorithm group would accrue the same knowledge acquisition and retention increases as the sequential group but would spend less time in total interacting with the Blank Slate application.

## 2. Materials and Methods

### 2.1. Participants

This randomized controlled trial lasted four weeks and included a total of 93 students from six different universities or colleges (Boston University, MA, USA; Campbell University, NC, USA; Chemeketa Community College, OR, USA; Pace University, NY, USA; Quinnipiac University, CT, USA; and Radford University, VA, USA) in the United States recruited through social media or email invitations. Approximately half were medical students, one-quarter community college students, and the remainder undergraduate students. Prior to participation, informed consent was obtained from all students and this study was approved by Quinnipiac University's Institutional Review Board.

### 2.2. Study Design

A three-arm parallel-groups pre-test-post-test study design with a 2:2:1 block randomization scheme was used to assign the 93 participants to either a control group (n = 19), a sequential group (n = 37), or an algorithm group (n = 37) (Figure 2). The distribution of medical students, community college students, and undergraduate students was similar across the three groups. This parallel-groups design had an overall power (i.e., overall F test) of 0.93 to detect differences between groups at the trial's post-test conclusion. This power analysis assumed that sequential (non-spaced) and algorithm (spaced) retrieval would have a large effect compared to no review (Cohen's d = 0.8).

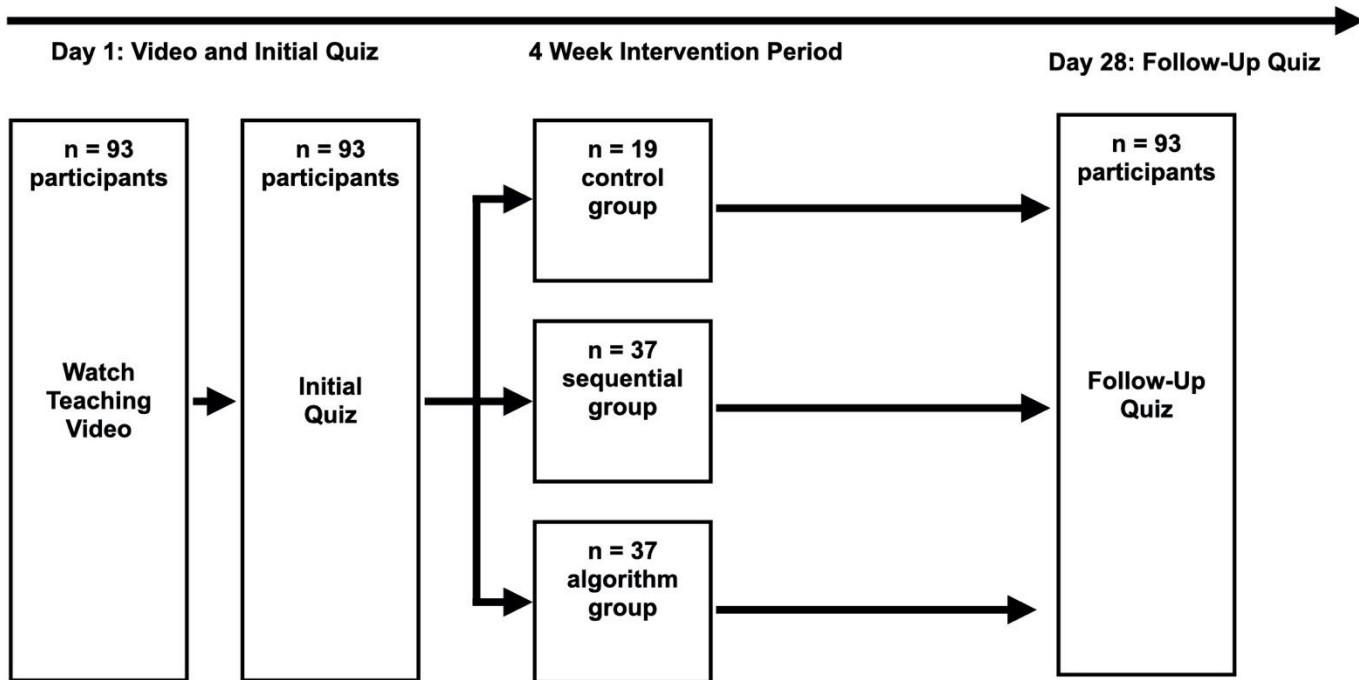

**Figure 2.** The randomized controlled trial's parallel-groups study design.

### 2.3. Intervention

All participants first viewed a 23 min teaching video on the Republic of Georgia and then immediately afterwards took a 60 multiple-choice-question (MCQ) quiz (pre-intervention). The teaching video was created by author J.M. using the Central Intelligence Agency's world factbook on Georgia [33] and was hosted by author D.M. via Zoom video conferencing (Zoom Video Communications, Inc; San Jose, CA, USA). Georgia was chosen in order to minimize prior knowledge as a confounder. The 60 MCQs were created by authors D.M., R.F., J.M., and M.T., and mapped to either the remember, understand, apply, or analyze tiers of Bloom's cognitive taxonomy [34], and were double Downing & Haladyna (2004)'s recommendation of 30 items for a test to adequately sample domain knowledge (i.e., knowledge of a specific, specialized field; in this case, Georgia) [35,36]. The questions asked about Georgia's geography, people and culture, government, economy, transportation, military, transnational issues, history, current affairs, education, and tourism (Table S1). Next, the sequential and algorithm groups were given access for four weeks to a Blank Slate account (Blank Slate Technologies LLC; Arlington, VA, USA) with 60 digital cards representing information tested in the MCQ quiz. The algorithm group experienced the authentic Blank Slate spaced retrieval algorithm, which selects the next card to be viewed based on previous individual performance; they reviewed however many cards were presented to them by the algorithm on a daily basis. The sequential group reviewed all 60 cards, presented in the same sequence, every day; the spaced retrieval algorithm was

disabled for this group. The control group had no access to Blank Slate. After four weeks, all participants took the same quiz again (post-intervention).

### 2.4. Data Analysis

Descriptive statistics include means with standard deviations for quantitative variables and frequencies with percentages for qualitative variables. An analysis of variance with Bonferroni-corrected post-hoc comparisons was used to compare pre- and post-intervention quiz scores between the three assigned groups. Paired t-tests were used to compare pre- to post-intervention scores within the same groups. Unpaired t-tests were used to compare mean total review time between the sequential and algorithm groups. These were chosen because the sample size was large enough (n = 74 for the 2 group comparisons) for the central limit theorem to apply, which states the sampling distribution of the sample means approximates normality and thus the t-test for group comparisons is acceptable. Moreover, Levene's test for equality of variance indicated no significant departure from the equal variance assumption and the equal number of students per group makes the t-test robust against test assumptions. To alleviate any potential concerns, non-parametric Mann–Whitney U tests were also conducted and the results were the same as the t-tests. Line graphs were used to display the daily averages for the Blank Slate account data collected from the sequential and algorithm groups over the four-week intervention period. For analyses, the app data were aggregated across days for each student resulting in an average value for each student and independent samples t-tests were performed to compare the sequential and algorithm groups. Analyses were conducted in SPSS v26 (IBM SPSS Statistics, Armonk, NY, USA) and the alpha level for statistical significance was set at 0.05.

### 3. Results

### 3.1. Pre- and Post-Intervention Quiz Scores

Figure 3 shows the pre- and post-intervention quiz scores for the control (no retrieval), sequential (non-spaced retrieval) and algorithm (spaced retrieval) groups. There were no significant difference between the pre-intervention scores ($p = 0.247$); which were 40 ($\pm6$), 42 ($\pm7$), and 43 ($\pm8$) questions correct for control, sequential, algorithm groups respectively. At post-intervention there was a statistically significant group difference between the scores ($p < 0.001$); which were 34 ($\pm5$), 59 ($\pm1$), and 58 ($\pm2$) questions correct for control, sequential, algorithm groups respectively. The control post-intervention mean score was significantly lower than the sequential ($p < 0.01$) and algorithm ($p < 0.01$) groups; but the sequential group did not differ from the algorithm group ($p = 0.274$). Pairwise contrasts revealed the control group's post-intervention mean score of 34 ($\pm5$) was significantly lower than their pre-intervention score of 40 ($\pm6$) ($p < 0.001$). The sequential group's post-intervention mean score of 59 ($\pm1$) was significantly higher than their pre-intervention score of 42 ($\pm7$) ($p < 0.001$). The algorithm group's post-intervention mean score of 57 ($\pm2$) was significantly higher than their pre-intervention score of 43 ($\pm8$) ($p < 0.001$).

Figure 4 shows a scatterplot of each individual participant's relative delta (i.e., the difference between their pre- and post-intervention score) by group. The range of delta scores was $-1$ to $-15$, 6 to 29, and 0 to 33 for the control, sequential, and algorithm groups, respectively.

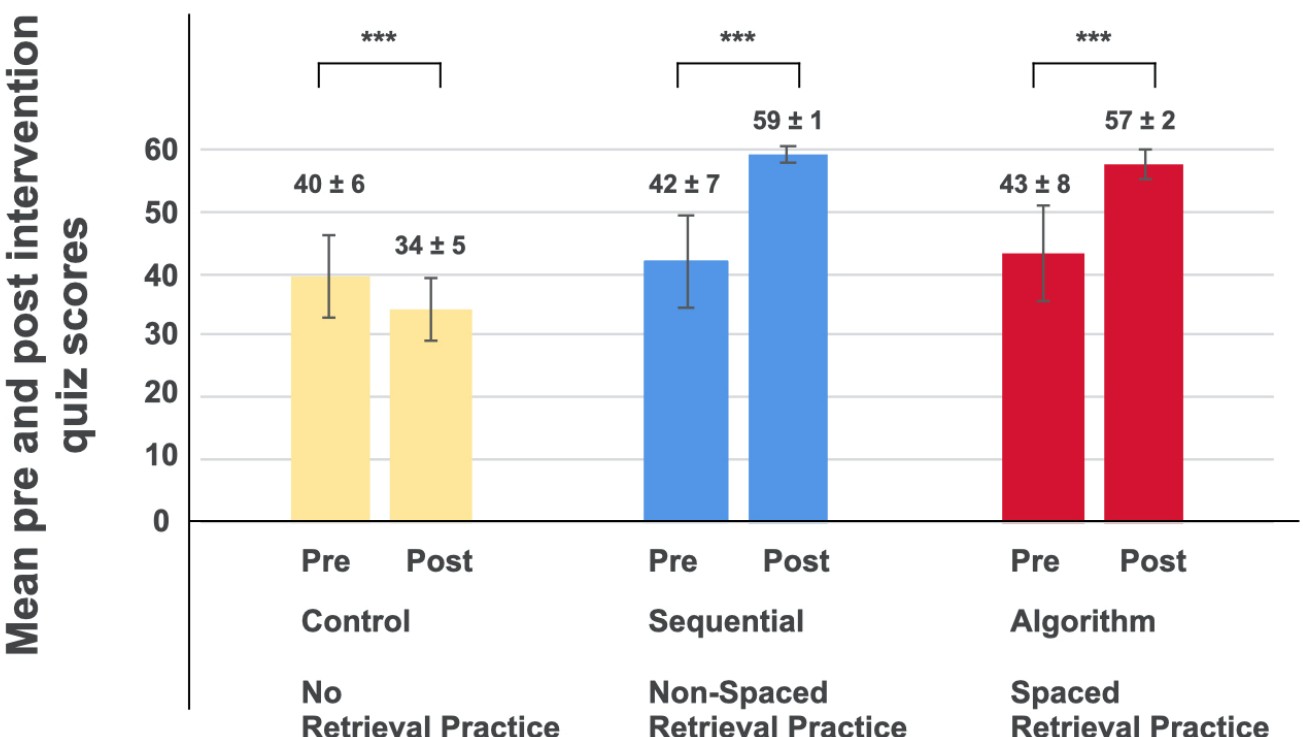

**Figure 3.** Mean (±standard deviation) pre- and post-intervention quiz scores for the control, sequential, and algorithm groups; *** = $p < 0.001$.

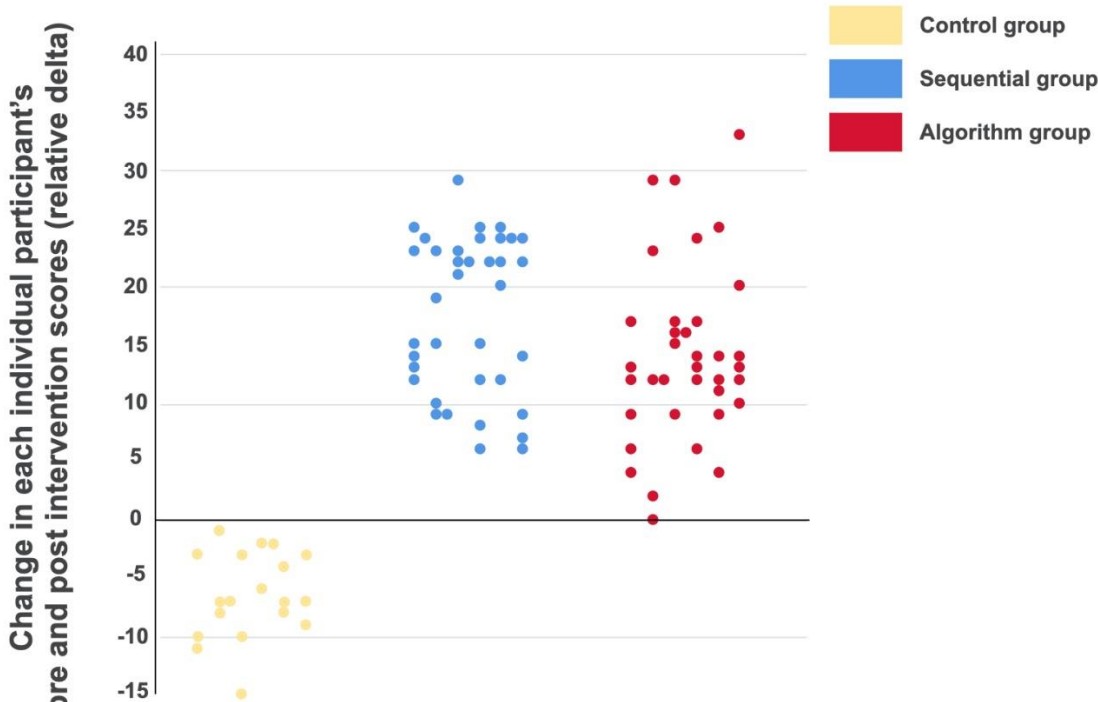

**Figure 4.** Scatterplot of each individual participant's relative delta (i.e., the difference between their pre- and post-intervention score) by group.

Figure 5 shows the mean number of questions answered correctly by each group at the start and end of the four-week intervention, as well as their mean review time. The control group's mean number of questions answered correctly decreased by 6 (±4). The

sequential group's mean number of questions answered correctly increased by 18 ($\pm$7). The algorithm group's mean number of questions answered correctly increased by 14 ($\pm$8). The sequential group spent a mean of 96 ($\pm$46) mins in total reviewing the 60 digital cards presented to them by Blank Slate; in contrast, the algorithm group spent a mean of 33 ($\pm$11) mins. This ~66% reduction of mean review time was significantly different ($p$ = 0.025).

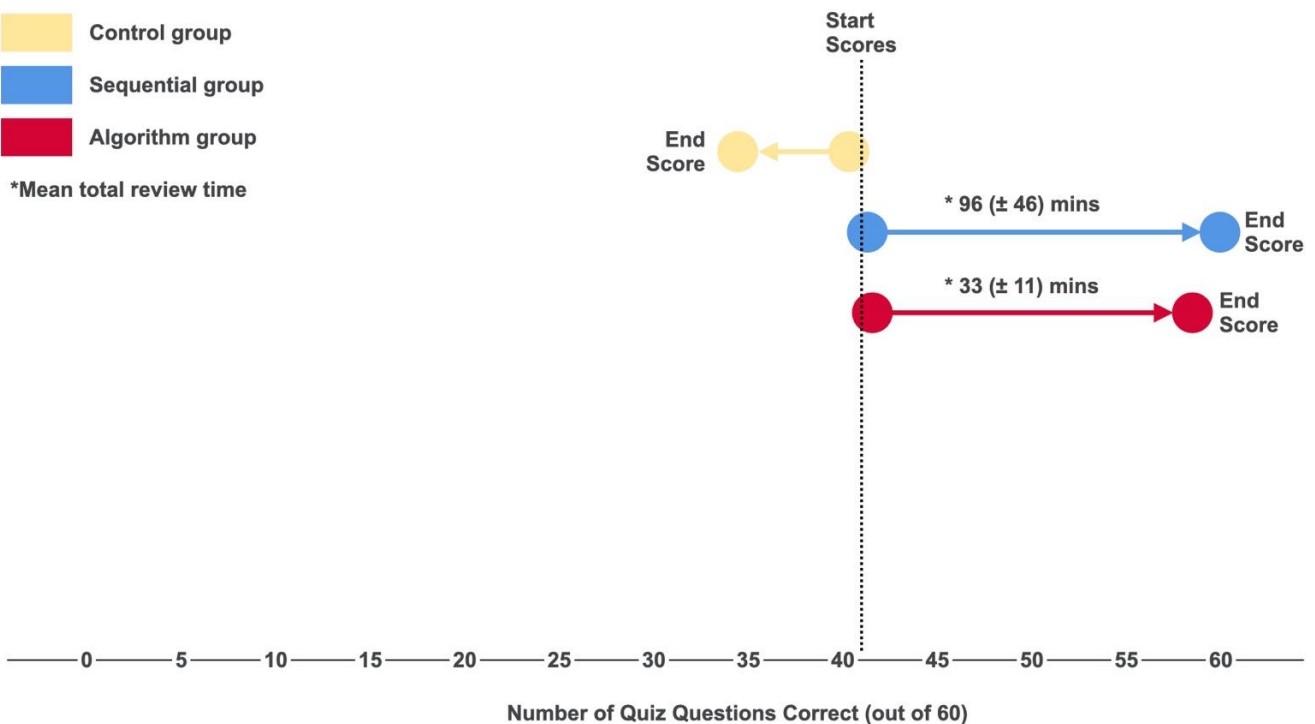

**Figure 5.** Mean number of questions answered correctly by each group at the start and end of the four-week intervention, and the mean total review time for the sequential and algorithm groups.

### 3.2. Comparison of Retrieval-Related Usage Characteristics

Figure 6 shows the mean number of digital card-prompted retrieval attempts each day by group. There were two students in the sequential group who repeatedly cycled through the deck of digital cards again and again on the first day (over 10 times) and this inflated the mean. Excluding the first day, the sequential group had a mean of 62.7 ($\pm$17.2) retrieval attempts per day compared to 16.3 ($\pm$11.2) for the algorithm group, which was consistent with the non-spaced retrieval and spaced retrieval underlying design for these two arms of the trial. This amounted to almost four times as many required retrieval attempts per day compared to the algorithm group; a difference that was significant and very large ($p$ < 0.001, d = 3.27, Table 1).

The sequential group successfully retrieved from long-term memory a slightly greater amount of information relevant to the 60 digital Blank Slate cards than the algorithm group; but this difference of 90.7% ($\pm$15.7) versus 88.4% ($\pm$6.4) was not significant and the size of the difference was small ($p$ = 0.704, d = 0.21, Table 1).

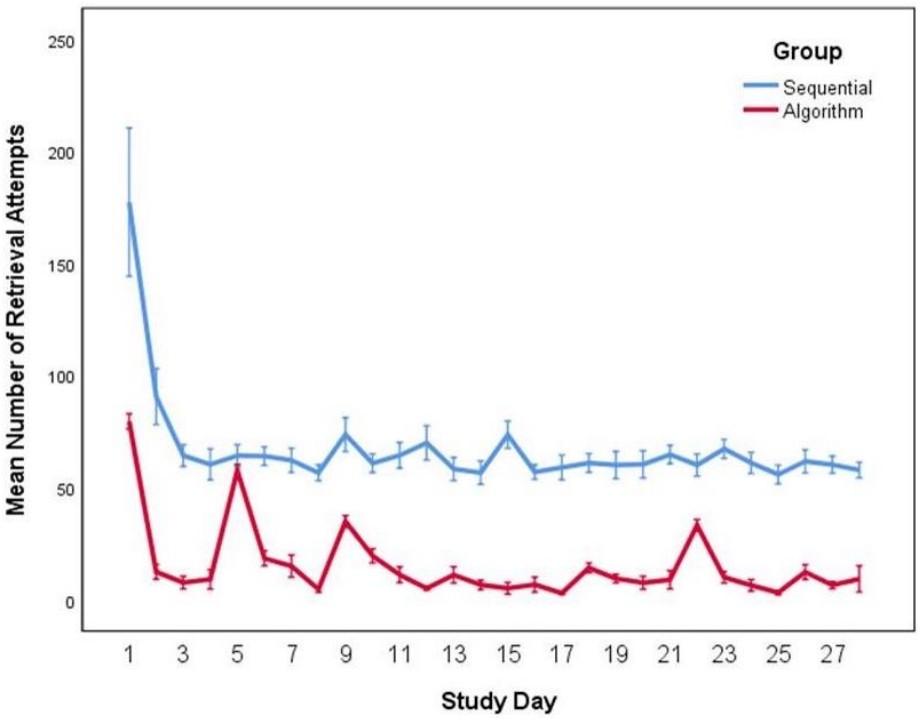

**Figure 6.** Mean (±standard error of the mean) number of digital card-prompted retrieval attempts each day by the sequential and algorithm groups.

**Table 1.** Mean (±standard deviation) of Blank Slate outcomes by group; * excludes study day 1.

| Outcome | Sequential | Algorithm | *p*-Value | Cohen's d |
|---|---|---|---|---|
| Retrieval attempts per day * | 62.7 (±17.2) | 16.3 (±11.2) | <0.001 | 3.27 |
| Percentage of successful retrievals from memory | 90.7 (±15.7) | 88.4 (±6.4) | 0.704 | 0.21 |
| Seconds per retrieval attempt | 4.3 (±1.7) | 6.7 (±2.5) | <0.001 | −1.14 |
| Minutes per day using Blank Slate * | 4.2 (±1.6) | 1.8 (±2.1) | <0.001 | 1.23 |

Figure 7 shows the mean time in seconds spent per retrieval attempt by the sequential and algorithm groups. There was no difference between groups for the first day of Blank Slate use when both groups were expected to review all 60 cards. However, by the second day, there was a noticeable difference, where the algorithm group was spending more time retrieving information pertaining to the digital cards presented to them. Across days, the algorithm group spent over 50% longer ($6.7 \pm 2.5$ s vs $4.2 \pm 1.7$ s) retrieving information per card and this difference was significant and large ($p < 0.001$, d = −1.14, Table 1). Although the sequential group took less time reviewing each question, this did not compensate for the more questions they reviewed and thus overall the sequential group spent more time each day reviewing cards (Figure 8 and Table 1). The algorithm group was more efficient and spent less than half as much time each day reviewing cards ($1.8 \pm 2.1$ min vs $4.2 \pm 1.6$ min) and the difference was significant and large ($p < 0.001$, d = 1.23, Table 1).

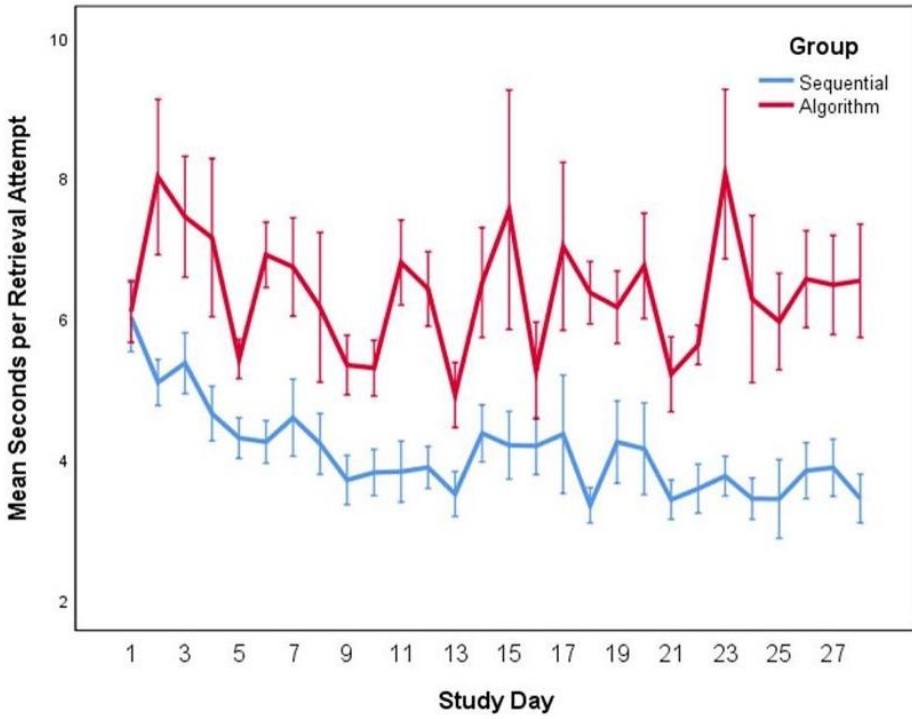

**Figure 7.** Mean (±standard error of the mean) time in seconds spent per retrieval attempt each day by the sequential and algorithm groups.

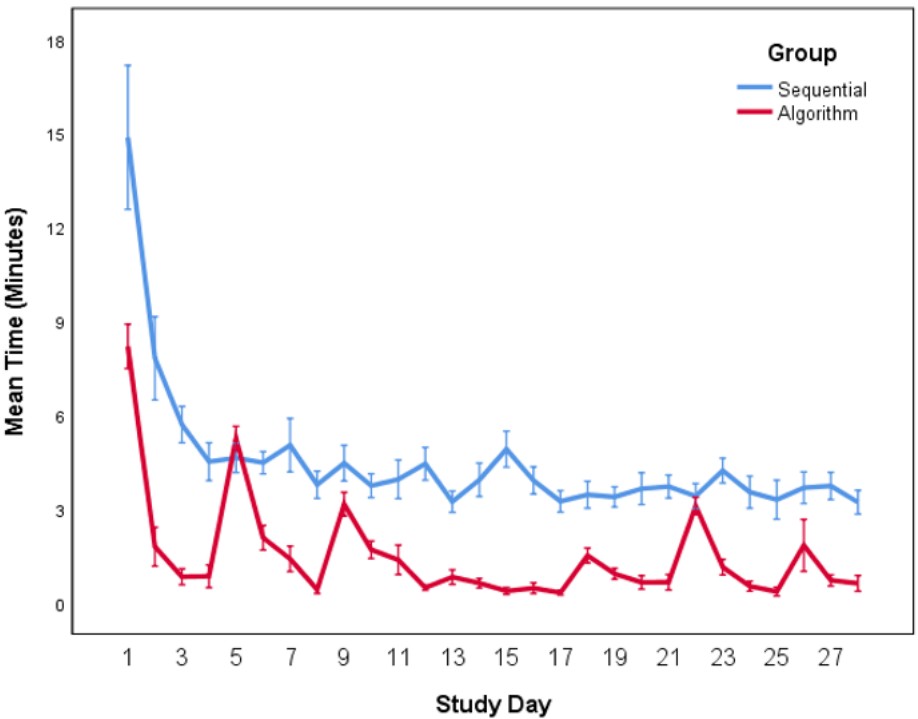

**Figure 8.** Mean (±standard error of the mean) time in minutes spent reviewing Blank Slate's digital cards and retrieving information each day by the sequential and algorithm groups.

Table 2 and Figure 9 shows the distribution of digital card-prompted retrieval attempts coded as easy (i.e., successful retrieval from memory was effortless), hard (i.e., successful retrieval from memory was difficult), or forgot (i.e., retrieval from memory was unsuccessful) by participants in the sequential and algorithm groups. These data help explain why the algorithm group exhibited more retrieval time per card (Figure 6). As the 'All

Viewings' row of Table 2 and first column of Figure 9 shows, during the four weeks of use, the sequential group reviewed a higher proportion of digital cards that prompted retrieval attempts they deemed easy, while the algorithm group reviewed a higher proportion that prompted retrieval attempts they deemed hard or as retrieval failures (i.e., forgot). The Blank Slate spaced retrieval algorithm prioritizes the selection of digital cards coded as hard or forgot by individual users for subsequent retrieval practice. It is unsurprising that digital cards coded previously as hard or forgot would prompt retrieval attempts that take longer when they are encountered again.

**Table 2.** Total number of digital card viewings and percentage of retrieval attempts coded as easy, hard, or forgot by participants in the sequential and algorithm groups for their first, third, sixth, and tenth viewings.

|  | Easy | Hard | Forgot |
|---|---|---|---|
| **All Viewings** | | | |
| Sequential | 44,053 (87.8%) | 1596 (3.2%) | 4512 (9.0%) |
| Algorithm | 7286 (72.4%) | 1034 (10.3%) | 1743 (17.3%) |
| **1st Viewing** | | | |
| Sequential | 57.7% | 9.6% | 32.7% |
| Algorithm | 61.6% | 7.7% | 30.6% |
| **3rd Viewing** | | | |
| Sequential | 83.6% | 8.5% | 7.9% |
| Algorithm | 79.9% | 7.3% | 12.8% |
| **6th Viewing** | | | |
| Sequential | 88.0% | 3.6% | 8.4% |
| Algorithm | 76.0% | 12.1% | 11.9% |
| **10th Viewing** | | | |
| Sequential | 91.4% | 2.3% | 6.3% |
| Algorithm | 75.1% | 10.7% | 14.2% |

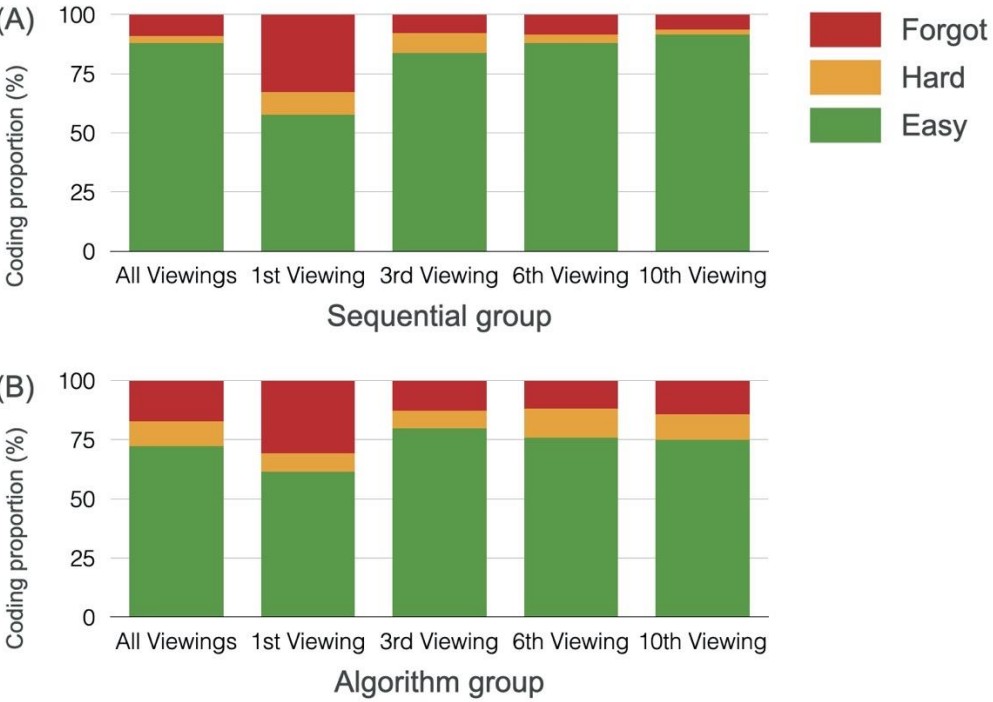

**Figure 9.** Percentage proportion of retrieval attempts coded as easy, hard, or forgot by participants in the (**A**) sequential and (**B**) algorithm groups for all viewings then their first, third, sixth, and tenth viewings.

The first time a digital card is viewed, there is no information on its retrieval difficulty level and so the distribution of easy, hard, and forgot is similar between the sequential and algorithm groups ('1st Viewing' row of Table 2). By the third viewing, the sequential group is encountering proportionally more easy questions ('3rd Viewing' row of Table 2). By the sixth viewing, the algorithm group is being presented with proportionally more hard and forgot cards ('6th Viewing' row of Table 2). By the tenth viewing, there is a large and clear difference, with the algorithm group being over four times as likely to be presented with a hard card and over twice as likely to be presented with a previously forgotten card ('10th Viewing' row of Table 2). This cannot be explained as better performance by the sequential group since there was no difference in the percentage of successful retrievals (Table 1).

### 3.3. Learner Analytics

Blank Slate's Total Knowledge Analytics$^{\text{TM}}$ platform provided a rich array of computationally analyzed data reports pertaining to each individual learner and the 74 participants (from the sequential and algorithm groups combined) as a small population. Illustrative excerpts for selected learner analytics reports are shown; learner names and email addresses have been replaced with pseudonyms to maintain their anonymity.

Figure 10 shows leaderboard rankings organized by either (A) user participation (i.e., the cumulative number of sessions a user has logged into so as to interact with Blank Slate, and (B) user accuracy (i.e., the latest proportion of successful retrieval attempts expressed as a percentage. These reports can display leaderboard statistics for a given week, month or for the entire history of Blank Slate use.

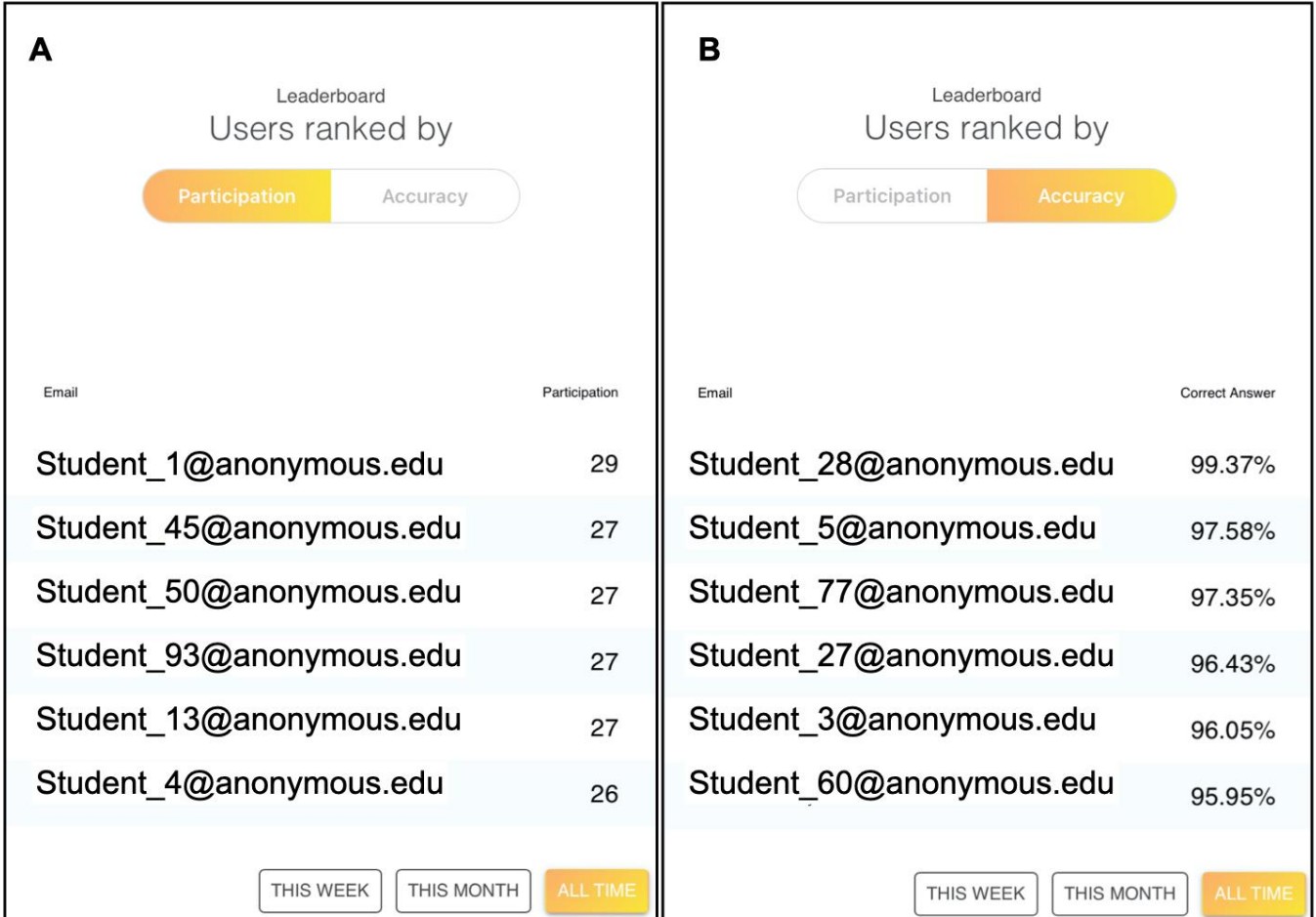

**Figure 10.** Blank Slate learner analytics—a screenshot of the leaderboard of anonymized participants ranked by (**A**) participation and by (**B**) accuracy.

Figure 11 displays the six participants who had the worst retrieval performance. The ranking may be reversed to display participants in order of best-to-worst performance. This screenshot is from the earliest days of the four-week trial when 'Student_31@anonymous.edu' was the participant who struggled the most to retrieve the information prompted by the digital cards.

| User email | Total sessions | Correct sessions | Total answer | Rank |
|---|---|---|---|---|
| Student_31@anonymous.edu | 27 | 186 | 1620 | 11% |
| Student_88@anonymous.edu | 20 | 292 | 411 | 71% |
| Student_71@anonymous.edu | 27 | 322 | 439 | 73% |
| Student_35@anonymous.edu | 29 | 328 | 447 | 73% |
| Student_6@anonymous.edu | 13 | 248 | 318 | 78% |
| Student_50@anonymous.edu | 18 | 281 | 357 | 79% |

**Figure 11.** Blank Slate learner analytics—a subset of anonymized participants ranked by worst to best performance.

Figure 12 shows (in yellow) the amount of time learner 'Student_42@anonymous.edu' spent engaging in retrieval practice each day from October 6, 2020 to November 1, 2020. Figure 12 also shows (in green) this participant's learning progress over time. On October 6, their retrieval from memory attempts were ~60% successful. This climbed to 100% success on October 22, October 24, and October 29 through November 1, 2020.

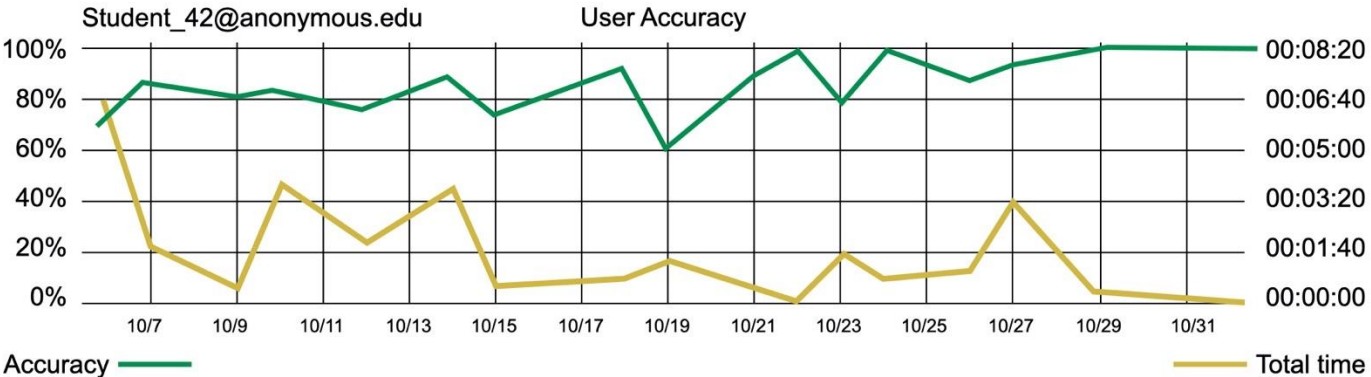

**Figure 12.** Blank Slate learner analytics—a single anonymized participant's total time spent interacting with Blank Slate (yellow) each day of the four-week trial and their percentage of successful retrievals from memory (green). Lefthand y axis: percentage of successful retrievals from memory. Righthand y axis: time in minutes and seconds spent interacting with Blank Slate. X axis: days from October 6 2020 to November 1 2020 using the U.S. date format (mm-dd); for example, 10/7 = October 7 2020.

Figure 13 displays 'Student_42@anonymous.edu''s coding of the 60 digital cards that prompted spaced retrieval as easy (green), hard (orange), and forgot (red) using a pie chart to indicate proportional breakdown over the four-week trial. This participant experienced 323 retrieval attempts—194 they coded as easy (60.1%), 100 as hard (30.9%), and 29 as forgot (i.e., cannot recall) (9.0%).

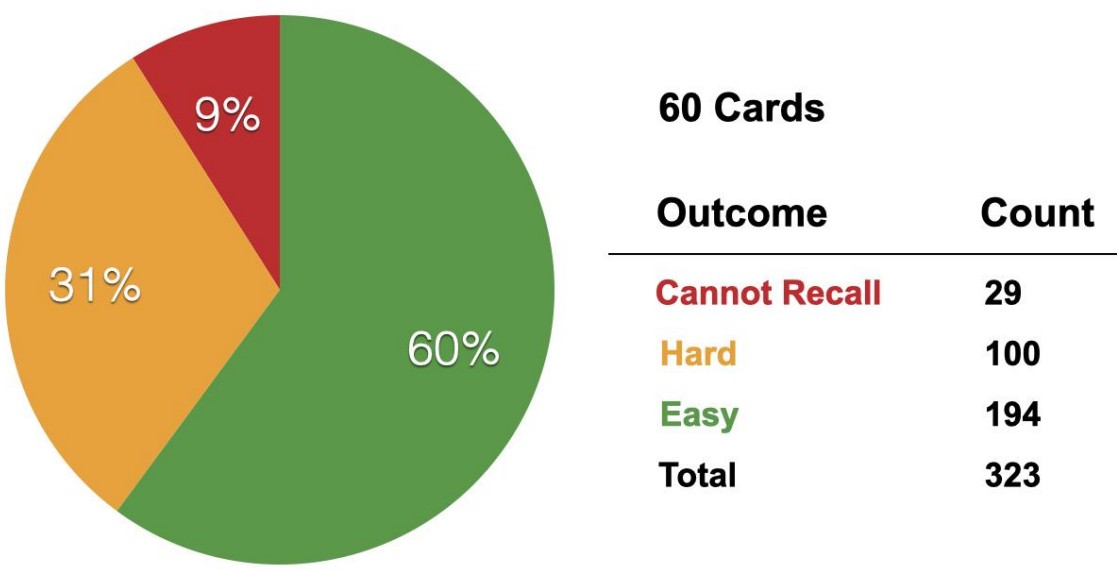

**Figure 13.** Blank Slate learner analytics—a single anonymized participant's coding of the 60 digital cards that prompt spaced retrieval as easy (green), hard (orange), or forgot (i.e., cannot recall) (red).

Figure 14 displays a selection of the 60 digital cards that 'Student_42@anonymous.edu' coded as easy, hard, or forgot (i.e., cannot recall). These data could form the basis for tailored feedback to the learner (for self-regulated learning) or educator (for coaching).

Student_42@anonymous.edu

| Date | Deck Name | Card Name | Answer |
|---|---|---|---|
| 2020-11-14 20:29:43 | Georgia Study Deck | Which country is Georgia's northern neighbor? | Easy |
| 2020-11-14 20:29:25 | Georgia Study Deck | How long is the approximate travel time between Tsibili and Batumi by rail? | Cannot Recall |
| 2020-11-14 20:29:09 | Georgia Study Deck | Economic growth was fastest during which years? | Hard |
| 2020-11-14 20:28:48 | Georgia Study Deck | What is the capital of Georgia? | Easy |
| 2020-11-14 20:28:41 | Georgia Study Deck | True or False: Georgia is slightly smaller than Ireland. | Easy |
| 2020-11-14 20:28:31 | Georgia Study Deck | Russia supports separatist movements in Georgia. Which two areas do they focus on? Select two answers. | Easy |
| 2020-11-14 20:28:25 | Georgia Study Deck | True or False: Georgia is slightly smaller than Ireland. | Cannot Recall |
| 2020-11-14 20:28:16 | Georgia Study Deck | Russia supports separatist movements in Georgia. Which two areas do they focus on? Select two answers. | Cannot Recall |
| 2020-11-14 20:28:01 | Georgia Study Deck | The August 2008 conflict with Russia affected GDP growth in the following years - by about how much? | Easy |
| 2020-11-14 20:27:10 | Georgia Study Deck | True or False: Georgia is often used by ciminal organizations and others as a transit for opiates in particular from Central Asia to Europe | Easy |

**Figure 14.** Blank Slate learner analytics—an excerpt of specific digital cards that a single anonymized participant's coded as easy (green), hard (orange), or forgot (i.e., cannot recall) (red).

Figure 15 shows the bottom 8 items from a ranked organization of the 60 digital cards for 'Student_42@anonymous.edu'. They are ranked from hardest to easiest to retrieve the relevant information solicited from memory.

Student_42@anonymous.edu

| Deck | Card ID | Card Name | Correct Answers | Total Answers | Rank |
|------|---------|-----------|-----------------|---------------|------|
| Georgia Study Deck | 15854 | Russia supports separatist movements in Georgia. Which two areas do they focus on? Select two answers. | 6 | 9 | 67% |
| Georgia Study Deck | 15864 | True or False: A previous president, Mikhael Saakashvili, was associated with Russia more than the West. | 7 | 10 | 70% |
| Georgia Study Deck | 15897 | How long is the approximate travel time between Tsibili and Batumi by rail? | 7 | 10 | 70% |
| Georgia Study Deck | 15859 | The Georgian language is part of what language group? | 7 | 10 | 70% |
| Georgia Study Deck | 15852 | True or False: Georgia is slightly smaller than Ireland. | 5 | 7 | 71% |
| Georgia Study Deck | 15889 | Is you subtracted the depth of Georgia's second deepest cave from its deepest, what would the total resulting depth be? | 5 | 7 | 71% |
| Georgia Study Deck | 15900 | Economic growth was fastest during which years? | 5 | 7 | 71% |
| Georgia Study Deck | 15885 | In which mountain range is Georgia nested? | 5 | 7 | 71% |

**Figure 15.** Blank Slate learner analytics—an excerpt of the specific digital cards that a single anonymized participant found to be difficult to retrieve ranked by hardest to easiest.

## 4. Discussion

The assumption that teaching results in durable learning is foundational to many of our educational activities. Unfortunately, information is often lost soon after it is learned [37]. According to the trace theory of memory, neurochemical signaling encoding information evokes synaptic pattern alterations in the brain reflective of a memory 'trace'. Information in short-term working memory lasts several seconds and if it is not rehearsed the neurochemical signaling quickly fades, leaving the synaptic reorganizations that were newly forming weak and unstable [37]. Learning is fundamentally about retrieving; we learn in order to later remember and apply what we have recalled. Practicing retrieval while you study is thought to strengthen the neurosynaptic connections that are the basis of memory traces, enriching and improving the learning process [38]. Indeed, there has been an exponential rise in the number of papers published on retrieval practice in education journals between 1991 and 2015 [9]. Alongside this, technology enhanced learning has become well positioned as an interface between digital technology and higher education teaching to flatten the human forgetting curve by facilitating spaced retrieval practice. We investigated whether Blank Slate was able to prevent forgetting and instead promote learning, while gathering and analyzing data in the background that could give insights into learners' performance. The data collected support our three hypothesized outcomes.

As can be seen in Figures 3–5, Blank Slate prevented the forgetting experienced by the control group participants. Not only that, Blank Slate facilitated further knowledge acquisition and long-term retention through retrieval practice. This is consistent with research in the cognitive and educational psychology field which has shown that time spent retrieving information is much better than simply reading or reviewing the information for the equivalent amount of time [39]. People may consider this counterintuitive because their experience of studying was to repeatedly re-read or re-write information, or highlight a text, or make carefully arranged and colorful study notes. Repeatedly retrieving actually works much better for long-term retention. Retrieval practice also leads to better transfer to new forms of questions than does repeated reading, reviewing, or re-writing. For example, a practice question might be about sonar in submarines and how bouncing sounds off objects in the water localizes those objects. Then the question on the final exam might be about sonar in bats—the same principle is at work, but in a different context [40,41]. Furthermore, retrieval practice leads to greater learning than educational techniques like concept mapping that are intended to nurture deep comprehension [10]. Blank Slate's algorithm produced authentic spaced retrieval in the algorithm group. Figure 6 shows (after

some initial enthusiasm in the first two days by participants in the sequential group that led them to have >60 retrieval attempts) that the sequential group interacted with all 60 digital cards across the four-week trial. Even as proportionally these retrievals became easier as Table 2 shows: *easy* was 57.7% (1st viewing), 83.6% (3rd viewing), 88.0% (6th viewing), and 91.4% (10th viewing). For the algorithm group, Figure 6 shows a much lower daily retrieval burden, with some spikes at days 5, 9, and 22, and even then those spikes were smaller than the baseline of the sequential group. To sum up, participants in the algorithm (spaced retrieval) group experienced the same learning improvements as the sequential (non-spaced retrieval) group with fewer retrieval attempts and less time consumed. Time is precious. Therefore, another advantage of spaced retrieval over non-spaced daily practice is that the time freed up can be used for other activities.

In today's environment, learning generally occurs in one of two contexts: synchronous or asynchronous. Students who take part in synchronous learning (i.e., happening with others at the same time) have to reserve time and commit to a specific meeting in order to attend live teaching sessions or online courses in real time. This may not be ideal for those who already have busy or compressed schedules. Asynchronous learning (i.e., happening independently at many different times) on the other hand can occur even when the student or teacher are not contemporaneous. Students will typically complete learning activities on their own and merely use the internet as a support tool rather than venturing online solely for interactive classes. With technology enhanced learning, we now not only have the means to make asynchronous learning resources available, but also to unobtrusively monitor their use [42]. Blank Slate is an example of a software application that supports asynchronous learning and facilitates our ability to weave in background assessment in real time: we can ask people what they know, ask them to commit to short answers, then generate learning data both for individuals and whole groups of learners based on their responses. Such, digitally-enhanced assessments were defined by the International Summit on Information Technology in Education (EDUsummIT) 2013 as those that "integrate: (1) an authentic learning experience involving digital media with (2) embedded continuous unobtrusive measures of performance, learning and knowledge, (i.e., 'stealth assessment') which (3) creates a highly detailed data record that can be computationally analyzed and displayed so that (4) learners and teachers can immediately utilize the information to improve learning." [43]. Unobtrusive assessment is seamlessly woven into the fabric of Blank Slate's digital environment. Its learner analytics platform represents a subtle, yet powerful process by which learner performance data are continuously gathered during the course of learning to support inferences made about the level of progress towards content mastery [44]. Blank Slate's convergence of asynchronous technologies that embed unobtrusive assessments/analytics with spaced learning motifs may represent an exciting opportunity to further competency-based learning.

### 4.1. Implications for Teaching and Learning Practice

This investigation adds to the growing body of knowledge about spaced retrieval practice (for a comprehensive review see Karpicke, 2017 [9]). It reproduced, by means of Blank Slate, learning benefits ascribed to spaced retrieval in a sample of Higher Education learners located across the United States. It further contributes the verification of Blank Slate as a novel spaced retrieval software application with embedded real-time learner analytics. Blank Slate presents an opportunity for spaced retrieval practice and externally-sourced feedback to be brought into close proximity via a shared digital platform. This is significant because feedback coupled to retrieval practice dramatically amplifies knowledge retention improvements [4,45,46]. Indeed, giving learners feedback to guide their progress has always been viewed as beneficial to learning; however, the need to combine opportunities for learner reflection and coaching with feedback to help learners achieve anticipated outcomes has been under stressed [47]. That said, unguided learner reflection often lacks fidelity [48,49]. Eva and Regehr noted that traditional self-assessment by learners usually takes the form of a "personal, unguided reflection on performance for the purposes of

generating an individually-derived summary of one's own level of knowledge, skill, and understanding in a particular area." [50]. In other words, individuals intuitively see themselves are the best source of information and look inward to generate an assessment of their own knowledge and abilities. Blank Slate's learner analytics reports (see Figures 10–15 for examples) represent an external source of information to inform learner reflection, self-directed learning processes, and feedback shared by educators. It provides a digital means to direct our attention to trustworthy feedback and to share it in a way that may improve its acceptance by the recipients [51]. In a learner-centered coaching approach, teachers could use available performance data generated by Blank Slate to contribute specific feedback that is relevant to fostering learners' continued momentum towards competence and then mastery. In this way, it may also help educators shift practically away from using assessments only to gather evidence "of" learning (i.e., at the end of a course or period of studying) and towards incorporating assessments "for" learning as a teaching strategy [8].

*4.2. Strengths, Limitations, and Future Studies*

Strengths of this investigation included the robust randomized controlled trial design and multi-institutional sample; however, this study's generalizability may be limited to university and college populations. The modest sample size is a major limitation; however, we believed it to be prudent to conduct a small study with sufficient power to establish that Blank Slate produces the benefits of spaced retrieval documented in the literature with one of its potential audiences (i.e., higher education learners) before any larger, follow-up studies are pursued. Another limitation is that we could only observe the learner analytics data and not act on it to provide feedback or coaching. To do so would have introduced a source of confounding that could have skewed interpretation of the spaced retrieval data. Future studies should focus on examining (i) the efficacy of Blank Slate with a more diverse representation of humans, and (ii) the effects of Blank Slate's learning analytics being used as source of feedback for self-regulated learning and learning coupled to educator-led feedback or coaching opportunities.

**5. Conclusions**

Our findings indicate that Blank Slate prevented forgetting and instead promoted learning and knowledge retention in a time-efficient manner. There is a need for technology-assisted identification of learning gaps that serve as a source of data input to guide individualized, formative feedback, and mentoring or coaching opportunities. Blank Slate as a technology enhanced learning tool integrated: (i) spaced retrieval learning experiences involving digital media with (ii) embedded continuous unobtrusive measures of performance, learning and knowledge, which (iii) created a highly detailed data record that was computationally analyzed and displayed so that (iv) stakeholders could utilize the information for feedback and coaching. Blank Slate, as an internet-based software application that can operate on multiple devices, is positioned to contribute to emerging digital infrastructure that supports education and learning.

**Supplementary Materials:** The following are available online at https://www.mdpi.com/2227-7102/11/3/90/s1, Table S1: List of the 60 author-created multiple-choice questions used for pre-test and post-test assessment.

**Author Contributions:** Conceptualization, D.M. and R.F.; data curation, D.M.; formal analysis, D.M. and R.F.; funding acquisition, M.T.; investigation, D.M.; methodology, D.M., R.F. and J.M.; project administration, D.M. and R.F.; resources, D.M., R.F., J.M. and M.T.; software, M.T.; supervision, D.M.; visualization, D.M. and R.F.; writing—original draft, D.M. and R.F.; writing—review and editing, D.M., R.F., J.M. and M.T. All authors have read and agreed to the published version of this manuscript.

**Funding:** Funding for this project was provided by Blank Slate Technologies LLC.

**Institutional Review Board Statement:** This study was conducted according to the guidelines of the Declaration of Helsinki and approved by the Human Experimentation Committee/Institutional Review Board of Quinnipiac University (#10620; 15 September 2020).

**Informed Consent Statement:** Informed consent was obtained from all subjects involved in this study.

**Data Availability Statement:** The data presented in this study are available on request from the corresponding author. The data are not publicly available in order to maintain participant confidentiality per IRB protocol.

**Conflicts of Interest:** D.M., R.F. and J.M. declare no conflicts of interest. M.T. declares a conflict of interest on the basis that he is the founder and CEO of Blank Slate Technologies LLC. Blank Slate Technologies LLC as the funder of this study had no role in the design of this study; in the collection, analyses, or interpretation of data; in the writing of this manuscript; or in the decision to publish the results. All authors confirm that the reported research results and their interpretation are free from inappropriate influences, manipulation, or suppression.

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
