# Peer review of "A Random Controlled Trial to Examine the Efficacy of Blank Slate: A Novel Spaced Retrieval Tool with Real-Time Learning Analytics"

_education, doi:10.3390/educsci11030090_

Round 1

Reviewer 1 Report

A very well-written paper on the very interesting topic of knowledge retention. All sections are clear and well defined. However, I would appreciate a more elaborated paragraph on the wider application of the findings in teaching and learning practice in the introduction and conclusion sections, supported with relevant examples. Moreover, besides the research objectives authors could consider to clearly state the research hypothesis in the introduction and its answer at the conclusion section, for readers that are not very familiar with the terminology used in the paper.

Author Response

Please refer to comments addressed to Reviewer #1.

Reviewer 2 Report

After reviewing the manuscript entitled "A Random Controlled Trial to Examine the Efficacy of Blank Slate: A Novel Spaced Retrieval Tool with Real-time Learning Analytics" I must point out a number of aspects that should be taken into account for its acceptance:

1.- The abstract exceeds 200 words. You must summarise it.
2.- The number of citations is scarce (there must be at least 45), as well as being very old. Authors should include more studies from the last 5 years. 
3.- The sample is relatively small. This is a major limitation in this study.
4.- The images and tables should be improved in terms of presentation and quality, especially the figures from figure 9 onwards. The use of a black background does not make them easier to read.
5.- The instrument used should be specified more clearly and in greater detail. 
The use of statistical t-tests should be justified, given that they are parametric tests. I doubt that with the sample chosen I can make use of this type of statistics.
The authors should include the theoretical and practical implications of the study. This is highly recommended.

The manuscript requires substantial changes and explanations in certain sections before it is accepted for publication.

I recommend that the authors work on these aspects.

Author Response

Please refer to comments addressed to Reviewer #2.

Round 2

Reviewer 2 Report

Dear author, the requested changes are still not properly made.

1.- The abstract is 211 words long. It should be less than 200.
2.- The number of citations has increased, but they are not mostly from the last 5 years. Please check this.
3.- The instrument is still not explained. I need to know the statistical values of the instrument.
4.- The use of various statistical tests is still not justified in the document.

In short, the changes made by the authors are not sufficient. Please revisit these aspects.

Author Response

Please reference to Reviewer #2 responses.
